# Micromotors of MnO_2_ for the Recovery of Microplastics

**DOI:** 10.3390/mi15010141

**Published:** 2024-01-17

**Authors:** Oscar Cervantes, Claudia Valtierra-Montiel, Laura Sampedro-Plata, Norberto Casillas, Nieves Menendez, Pilar Herrasti

**Affiliations:** 1Department of Applied Physical Chemistry, Faculty of Sciences, Autonomous University of Madrid, Francisco Tomás y Valiente 7, 28049 Madrid, Spain; ocervantesa@gmail.com (O.C.); laura.sampedrop@estudiante.uam.es (L.S.-P.); nieves.menendez@uam.es (N.M.); 2Department of Chemistry, Center of Exact Sciences and Engineering (CUCEI), University of Guadalajara, Marcelino García Barragán 1421, Col. Olímpica, Guadalajara 44430, Jalisco, Mexico; norberto.casillas@academicos.udg.mx; 3Master’s Program in Nanomaterials Science and Technology, Natural and Exact Sciences Division, University of Guanajuato, Noria Alta S/N, Guanajuato 36050, Guanajuato, Mexico; cd.valtierramontiel@hotmail.com

**Keywords:** manganese oxide, polystyrene, micromotors, recovering

## Abstract

Plastics, primarily microplastics, are among the greatest pollutants in aquatic environments. Their removal and/or degradation in these environments are crucial to ensure an optimal future of these ecosystems. In this work, MnO_2_ particles were synthesized and characterized for the removal of polystyrene microplastics as a model. MnO_2_ catalyzes the peroxide reaction, resulting in the formation of oxygen bubbles that propel the pollutants to the surface, achieving removal efficiencies of up to 80%. To achieve this, hydrothermal synthesis was employed using various methods. Parameters such as MnO_2_, pH, microplastics, and H_2_O_2_ concentrations were varied to determine the optimal conditions for microplastics recovering. The ideal conditions for a low microplastic concentrations (10 mg L^−1^) are 0.2 g L^−1^ MnO_2_, 1.6% of H_2_O_2_ and 0.01 triton as a surfactant. In these conditions, the micromotors can recover approximately 80% of 300 nm sized polystyrene microplastic within 40 min.

## 1. Introduction

Environmental pollution, and its impact on our lives and those of future generations, is currently arousing significant interest and concern in today’s society. One of the largest sources of pollution is also one of the most used materials in our daily lives: plastics.

Plastics are synthetic polymeric materials primarily derived from petroleum. Their properties, such as hydrophobicity, corrosion resistance, light weight, chemical inertness, and durability, explain their widespread popularity [1]. Worldwide, annual plastic production has surged from 2 million tons in the 1950s to 367 million tons in 2020 [2]. Predictions indicate that the total mass of plastic debris accumulated in the ocean could reach approximately 250 million metric tons (Mt) by 2025, which is an order of magnitude higher than in 2010 [3]. The most used types of plastics include high-density polyethylene (HDPE), low-density polyethylene (LDPE), polyvinyl chloride (PVC), polystyrene (PS), polypropylene (PP), and polyethylene terephthalate (PET), accounting for 90% of the plastics produced worldwide [4].

Depending on their size, plastics are classified as macroplastics (McP), which have an average size greater than 25 mm; mesoplastics (MsP) with a size between 25 and 5 mm; and microplastics (MP), on which this work focuses, whose size may vary according to the literature, but it is established here that these plastics have a size between 5000 and 1 µm; and finally, those with a size smaller than 1 µm, nanoplastics (NP). It is precisely these last two groups that are the most difficult to recover due to their small size, and they accumulate in oceans and watery areas (around 8 million tons of MP per year) [5].

Microplastics (MPs) can be further categorized into primary and secondary MPs. Primary MPs are those specifically manufactured in this size range, which includes, for example, those found in cosmetic skin cleansing products. On the other hand, secondary microplastics result from the degradation of macroplastics (McP) and mesoplastics (MsP). The degradation of these plastics can involve processes such as weathering and aging. Additionally, once they reach the micrometer size range, both primary and secondary MPs can undergo further degradation processes, leading to modifications in some of their properties, such as color or density, which can result in unexpected physical and/or chemical effects on the natural environment.

Since most plastics are reusable materials, one might consider not only their degradation, but also their recovery and subsequent use. However, only about 9% of the world’s plastic waste is currently recycled [6]. As a result, numerous studies have been conducted on microplastic recovery methods, including the utilization of magnetic carbon nanotubes, which have demonstrated a high adsorption capacity for hydrophobic aromatic compounds, such as certain plastics. Another recent method is based on the use of aluminum (III) and iron (III) coagulant salts, which react with water in a basic environment to form metal hydroxide particles that do not dissolve and can be separated once they have settled. These particles are capable of binding with microplastics through complexation, modifying the polymer bonds [7].

Among the various methodologies and mechanisms listed in Table 1 for the recovery of microplastics, the use of micromotors has received relatively limited research attention. This process essentially involves the decomposition of hydrogen peroxide by a catalyst to generate oxygen, as described in Reaction (1) [8,9].
(1)$$2H_{2}O_{2}\overset{MnO_{2}}{\to }2H_{2}O+O_{2}$$

Initially, noble metal catalysts were employed, but their scarcity and cost prompted the development of alternative catalysts. Notably, manganese oxide is one of the catalysts used for this purpose. The production of oxygen bubbles propels the contaminants to the liquid’s surface, creating foam that can be collected, within which the microplastics are located. Additionally, the use of these types of catalysts can facilitate the degradation of plastics by generating radicals in the presence of peroxides or persulfate, known as the Fenton process, as indicated in Reactions (2) and (3).
(2)$${Mn}^{4+}+H_{2}O_{2}\to {Mn}^{3+}+{HO}_{2}^{••}+H^{+}$$
(3)$${Mn}^{3+}+H_{2}O_{2}\to {Mn}^{4+}+{HO}^{•}+{OH}^{-}$$

In the application of this technology for microplastic recovery, it is sometimes necessary to introduce a surfactant into the medium to aid in the encapsulation of the microplastics within the generated foam. Surfactants alter the surface tension between the microplastics and the medium, enhancing their affinity for water. The application of this methodology within an industrial process would be possible after the realization of much more studied and known processes for the recovery of organic matter, for example the use of activated carbon filters and ultrafiltration techniques based on membrane reactors, as well as methodologies such as coagulation or electrocoagulation. Subsequent to these techniques and with already very low quantities of microplastics, the use of this technique could allow the elimination of these in small size baths or in continuous usage.

In this study, various manganese oxide nanostructures were synthesized and employed for the recovery of synthetic microplastics with a size of approximately 300 nm in the laboratory. Parameters such as the microplastic quantity, structure, morphology, catalyst dosage, hydrogen peroxide concentration, surfactant, and pH were systematically varied. The measurement of total organic carbon (TOC) in the suspension was used as the response variable to determine the optimal conditions for microplastic recovery.

## 2. Materials and Methods

All reagents used in this work were used without any further purification. KMnO_4_ (>99%) and hydrochloric acid (HCl 37 wt.% in H_2_O) were obtained from Labkem, Spain. Na_2_S_2_O_8_ (>98%, PanReac AppliChem, Darmstadt, Germany). Triton X-100 (Sigma-Aldrich, St. Louis, MO, USA). Hydrogen peroxide solution, (H_2_O_2_ 30 wt.% in H_2_O, Honeywell, Offenbach, Germany). Titanium oxysulfate (TiOSO_4_ 27–31% H_2_SO_4_ basis) and styrene were obtained from Sigma-Aldrich, Schenelldorf, Germany.

### 2.1. Synthesis and Characterization of MnO_2_

Different MnO_2_ structures were synthesized using hydrothermal synthesis, and the specific conditions for each synthesis are provided below.

0.627 g of KMnO_4_ were dissolved in 56 mL of deionized water with continuous magnetic stirring. Once the solution is homogeneous, 1.4 mL of HCl was added. After 15 min the solution was transferred to a 100 mL autoclave and place it in an oven maintained at 80 °C for 12 h. This sample is referred to as S80 [18].The following synthesis was conducted by dissolving 0.363 g of KMnO_4_ in 80 mL of deionized water. After achieving homogeneity in the solution, 0.8 mL of HCl was added. Subsequently, the mixture was subjected to continuous magnetic stirring for 1 h, following which it was transferred to an autoclave and maintained at 150 °C for 12 h. This sample was designated as ‘S150’ [19].The synthesis of the sample named S210 was carried out by adding 45 mL of a 0.60 mol/L solution of MnSO_4_ dropwise, using a burette, to a beaker containing 28.2 mL of a 0.60 mol/L solution of KMnO_4_, while maintaining constant magnetic stirring. The mixture was stirred continuously for 30 min, after which it was transferred to an autoclave and kept in the oven at 210 °C for 12 h [20].

In all cases, the oven was configured with a ramp of 1.5 °C/min until the selected temperature was reached. After 12 h, the temperature was decreased at a ramp of 3 °C/min, and then it was allowed to stand for approximately 4 h until it reached room temperature. After this period, each of the synthesized particles was washed with water and ethanol, undergoing centrifugation at 5000 rpm for 3 min, with each process being repeated three times. Once the solid was separated and cleaned, the precipitate was dried in a vacuum oven at 40 °C for 24 h.

The structure of the particles was characterized using a Theta/2Theta Bruker D8 diffractometer (Bruker, Billerica, MA, USA), which was equipped with a primary monochromator and an ultrafast Lynxeye XE-T multichannel detector with Cu Kα radiation. Diffractograms were recorded in the 2θ range from 5° to 80°, and their profiles were analyzed using a Fullprof (version 7.70) suite based on the Rietveld method or by the PANalytical X’Pert High Score program (version 2.0a). Micrographs and energy-dispersive X-ray spectroscopy (EDX) analysis were obtained through scanning electron microscopy (SEM) using a Hitachi S-3000N microscope (Hitachi, Tokyo, Japan).

The porous structure of the materials, which had been previously outgassed overnight at 150 °C to a residual pressure of <10^−3^ Torr, was characterized by nitrogen adsorption–desorption using a Micromeritics Tristar 3020 system (Micrometrics, Norcross, GA, USA).

Different MnO_2_ structures have been synthesized using a hydrothermal synthesis method. The specific conditions for each synthesis are detailed below.

### 2.2. Synthesis of Polystyrene

The synthesis of polystyrene particles was carried out following the methodology previously reported by Lu et al. [21]. Briefly, 150 mL of distilled water and 10 mL of styrene were placed in a round-bottom flask with a condenser immersed in a thermal bath at 80 °C. They were stirred for 30 min under an inert atmosphere of N_2_ gas. Subsequently, 5 mL of a 26.4 g L^−1^.

A Na_2_S_2_O_8_ solution was added dropwise with the aid of a syringe. Finally, the reaction was allowed to proceed for 6 h. The resulting suspended particles were thoroughly washed with distilled water and centrifuged at 11,000 rpm repeatedly. Once the microplastics were resuspended, the resulting suspension’s concentration was calculated using 0.22 μm Millipore filters. A known volume, typically 1–2 mL, was passed through the filter to quantify the amount of microplastics deposited on it.

To analyze the size distribution using an eLINE Plus equipment from Raith GmbH Co. (Dortmund, Germany) by FESEM, several aliquots were taken and deposited in glass sample holders with the aid of a micropipette. The samples were then allowed to dry at room temperature. Afterward, the glass sample holders were placed on carbon tape affixed to a sample holder and coated with a 10 nm layer of Cr to enhance conductivity. The morphology and size distribution are showed in Appendix A

### 2.3. Experimental Setup and Conditions for the Recovery of Microplastics

A 250 mL container was used, into which 100 mL of a MP solution was added. These experiments were conducted with various parameters being varied, including pH, the amount of MnO_2_, hydrogen peroxide concentration, microplastics concentration, and surfactant concentration. Initial concentrations of MnO_2_ were set at 0.1, 0.2, and 0.3 g L^−1^, hydrogen peroxide at 1.6%, 3%, and 6%, and microplastic amounts at 10 ppm, 20 ppm, and 50 ppm. Triton concentrations were 0.001%, 0.005%, and 0.01%, and the pH was adjusted to 3, 7, and 9 using 1 M HCl and NaOH, respectively.

The foam generated for the decomposition of the H_2_O_2_ was removed continuously for a duration of 20 to 40 min, depending on the test conditions. At this point, the process was considered complete since foam generation had become negligible. From the lower part of the container, samples were taken from the suspension, and the total organic carbon (TOC) content was measured using a colorimeter, specifically the HACH DR900, through a purging method with the aid of a HACH DRB200 instrument (Düsseldorf, Germany) and TNT reagents. The TOC measurement in each experiment was compared to the initial total organic carbon content corresponding to the microplastic and Triton surfactant. The remaining peroxide content was quantified using UV-Vis titanium complexation with a Perkin Elmer Lambda 365 UV-Vis apparatus (Waltham, MA, USA) [22]. Following each assay, the solution had to be agitated at a temperature of 50 °C for approximately 12 h to eliminate excess peroxide, preventing interference with the TOC measurement. In the case of manganese oxide, to prevent interference, all the aliquots were diluted to a 4-fold concentration, and no interference in the measurement was observed.

## 3. Results and Discussion

### 3.1. Characterization of the MnO_2_ Synthesized

SEM micrographs of the different samples synthesized at various temperatures are depicted in Figure 1. Their morphologies are distinctly different, but all of them consist of nanowires whose size increases with higher synthesis temperatures. In the case of the sample synthesized at a lower temperature (S80), the nanocrystals agglomerate to form compact spheres of varying sizes (1–3 μm). Sample S210 is composed of well-defined prisms, each about 2–3 μm in length. At an intermediate temperature, sample S150 exhibits agglomerations of ribbon-like nanowires with smaller base and length. In all cases, EDX measurements of the samples confirm the presence of an oxygen-to-manganese ratio very close to 2.

The XRD patterns of the samples resulting from the hydrothermal process under various conditions and temperatures are presented in Figure 2. The diffractogram corresponding to sample S80 could not be fitted by Rietveld and is shown in Figure 2a analyzed by the PANalytical X’Pert High Score program. This structure, as shown by SEM, is a spherical structure but composed of filaments oriented which makes it impossible to fit, but diffraction peaks can be exclusively indexed as a δ-MnO_2_ (JCPDS 01-072-1982). Samples S150 and S210 could be fitted by Rietveld and the fitting are shown in Figure 2b,c. A best fit is obtained for sample S210 with lattice parameters of a = b = 4.4001 (1), c = 2.8739 (1), this can be identified as β-MnO_2_ (JCPDS 01-081-2261). In the case of sample S150, the fit presents worse refinement mainly due to showing several preferential directions together with nanostructure, which makes it difficult to obtain accurate values of the lattice parameters, those obtained with the fit are, a = b = 9.8299, c = 2.8739 (1), identified with α-MnO_2_ (JCPDS 44-0141) [23]. However, it is evident that all the samples correspond to pure tetragonal phase of MnO_2_. Additionally, the crystallinity of the samples increases with rising temperature, which is consistent with the SEM micrographs.

In addition, N_2_ physisorption was conducted to determine the specific surface area (S_BET_) value and pore size distribution. Appendix A shows the nitrogen adsorption–desorption curves where it can be observed that only the sample δ-MnO_2_ S80 presents according to the IUPAC classification [24] a-type IV isotherm with a H3-type hysteresis loop, indicating the presence of mesopores. The samples α-MnO_2_ S150 and β-MnO_2_ S210 present more clearly a type II structure. The values extracted from the isotherms of the three synthesized samples are presented in Table 2. In all cases, the S_BET_ is relatively low, ranging between 57 and 13 m^2^/g. However, it is observed that the specific surface area decreases as the synthesis temperature in the hydrothermal process increases. At higher temperatures, they exhibit non-porous materials with low pore densities.

### 3.2. Recovery of Microplastics

Although all three synthesized samples were tested for the recovery of microplastics in the same conditions: pH 9, 0.2 g L^−1^ MnO_2_, 1.6% H_2_O_2_ and 0.01% triton. The results obtained were a 75% recovering with the S80 sample, 25% with S210 and a 20% with S150. The results indicated that the sample obtained with the lowest temperature and the highest surface area yielded the best results. Therefore, the variables influencing the recovery process will be analyzed using the sample labeled S80.

The influence of pH was investigated using conditions with 1.6% H_2_O_2_, a microplastic concentration of 20 ppm, 0.2 g L^−1^ MnO_2_, and 0.01% Triton. Under these conditions, foam formation and, consequently, the collection of a certain amount of microplastics only occur at basic pH levels. pH values lower than 9, even after prolonged reaction times, remove a very minimal amount of microplastics. This effect can be explained by the suppression of bubble generation under acidic conditions (see Appendix A). Conversely, as pH increases, the catalytically generated microbubbles adsorb suspended microplastics and rise to the surface of the reaction solution, effectively removing them from the environment [8]. Further increasing the pH beyond 9 does not significantly enhance bubble generation or microplastic recovery, which is why this pH level was selected.

Once the optimal pH was determined, we varied the amount of MnO_2_ while keeping other factors constant. In Figure 3, the ratio between the TOC obtained after the experiment and the initial TOC is depicted for MnO_2_ concentrations of 0.1, 0.2, and 0.3 g L^−1^. It is evident that there is no statistically significant difference in the recovery rates when using different MnO_2_ concentrations within the studied range, especially in combination with low concentrations of H_2_O_2_. The determined recovery rate remained at 75%.

Based on the limited impact of MnO_2_ concentration on microplastic recovery, we selected a MnO_2_ concentration of 0.2 g L^−1^ for subsequent experiments. It was also verified that the presence of MnO_2_ particles had a negligible effect on TOC measurements. Although an increase in pH beyond 9 did not result in a significant increase in bubble generation and, consequently, microplastic coating, we opted to maintain a pH of 9 and varying other parameters. We observed that as the peroxide concentration in the medium increased, there was a considerable amount of unreacted peroxide left in the medium, Figure 4. This had a significant impact on the TOC measurements. The main reason that can affect the TOC measurement is the presence in the medium of radicals coming from the reaction between hydrogen peroxide and MnO_2_ (reactions (2) and (3)). These radicals can produce the degradation of the microplastics, which results in organic compounds in solution that are not recovered by the foam formed, and that interfere later in the measurement of TOC. The calibration curves for H_2_O_2_ are provided in Appendix A, and statistical calculation are shown in Appendix A.

The remaining amounts of peroxides for the experiments with 3% and 6% concentrations were found to be significantly high, substantially affecting the TOC measurement. To ensure accurate and reproducible measurements, the measurement protocol was modified to ensure that the peroxide concentration prior to TOC measurement was negligible, as detailed in the experimental section. Figure 5 presents the results obtained for the amount of unrecovered MP in the foam at different quantities of microplastics and peroxide.

It can be observed that an increase in peroxide concentration does not result in a higher recovery of microplastic. The Appendix A, provides information on the foam formation for the different concentrations of hydrogen peroxide used and for a time of 5 min. It can be seen that the foam formed for a high concentration of peroxide produces a large amount of foam so that its removal from the solution cannot be controlled. For small amounts of microplastics and low peroxide concentrations, the recovery rate exceeds 70%. This suggests that this procedure is effective at recovering microplastics at low concentrations, with minimal peroxide usage, as the peroxide is eliminated from the medium and converts into oxygen when 1.6% concentrations are employed.

## 4. Conclusions

In summary, these MnO_2_ micromotors act as catalysts in the production of oxygen and can remove over 75% of microplastics through entrainment in just 40 min. The optimal conditions for use should be investigated based on the quantity and type of microplastics to be recovered and the specific environmental conditions.

The results of the current work demonstrate the potential use of these or other micromotors for the removal of emerging pollutants, such as microplastics. The advantage of this methodology is that it utilizes materials that are low-cost, readily available, and can be fabricated on a large scale.

## Figures and Tables

**Figure 1 micromachines-15-00141-f001:**
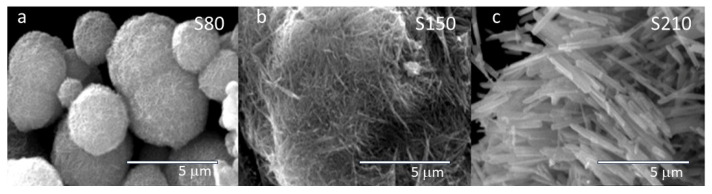
SEM micrograph of the samples obtained using different methodologies, labeled as (**a**) S80, (**b**) S150, and (**c**) S210, indicating the temperature of the hydrothermal method.

**Figure 2 micromachines-15-00141-f002:**
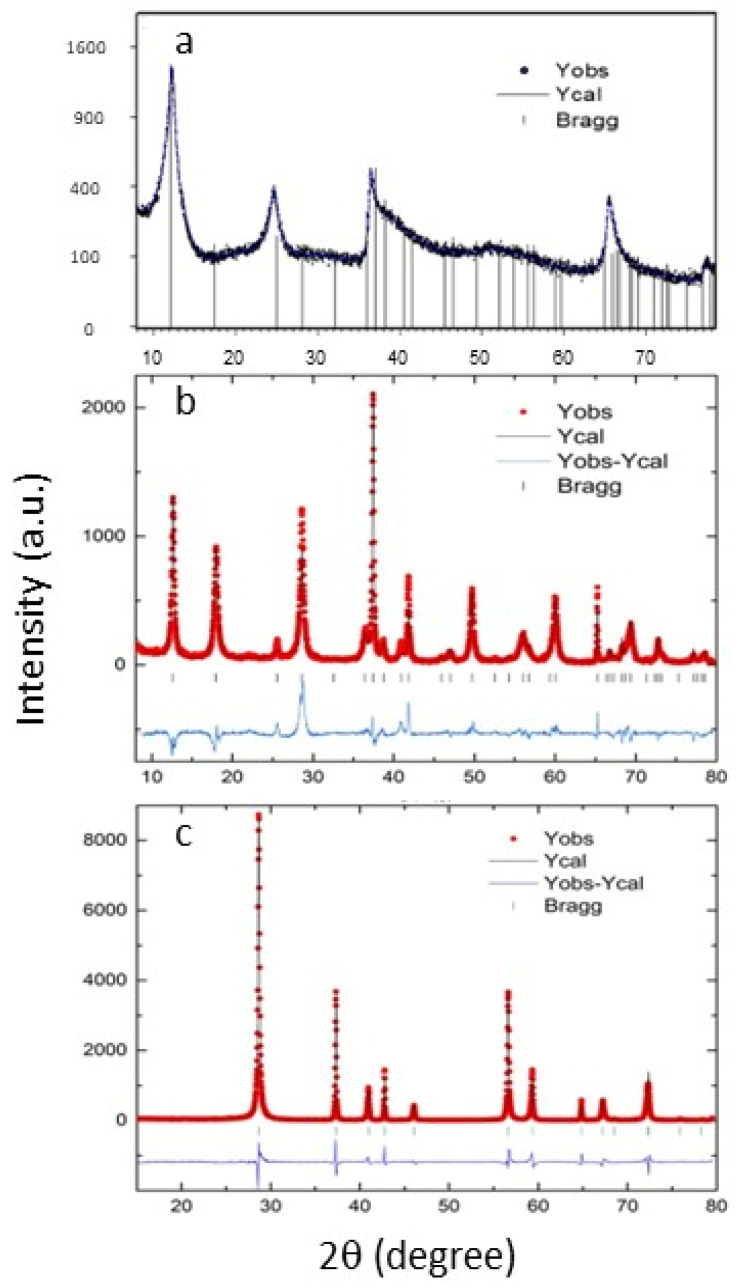
X-ray diffractogram, (**a**) Sample S80, (**b**) Sample S150 and (**c**) Sample S210. The observed, and calculated patterns are shown.

**Figure 3 micromachines-15-00141-f003:**
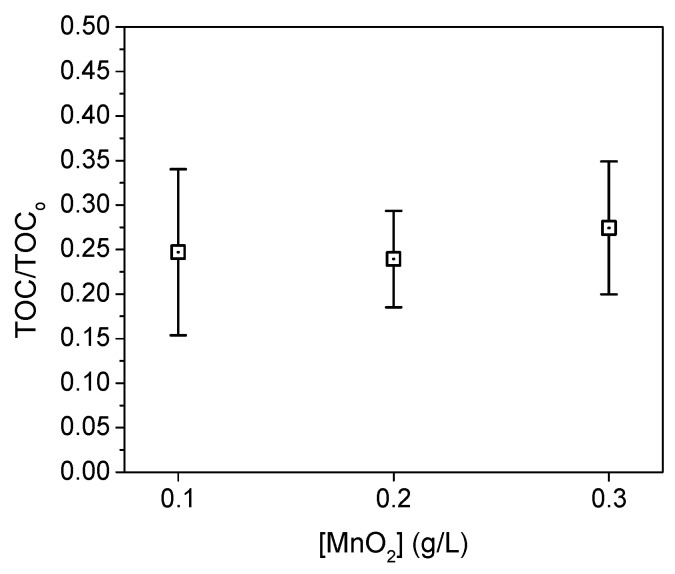
Means plot for the TOC/TOC_o_ vs. [MnO_2_]. 20 ppm PS, 1.6% H_2_O_2_, pH 9 and 0.01% Triton.

**Figure 4 micromachines-15-00141-f004:**
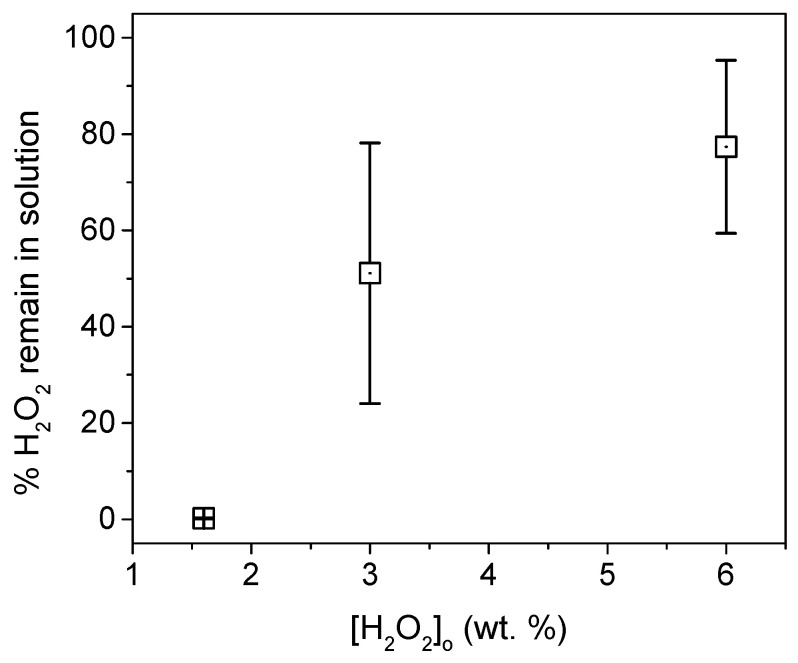
Amount of H_2_O_2_ remained in solution after 40 min of reaction with multiples eliminations of the foam.

**Figure 5 micromachines-15-00141-f005:**
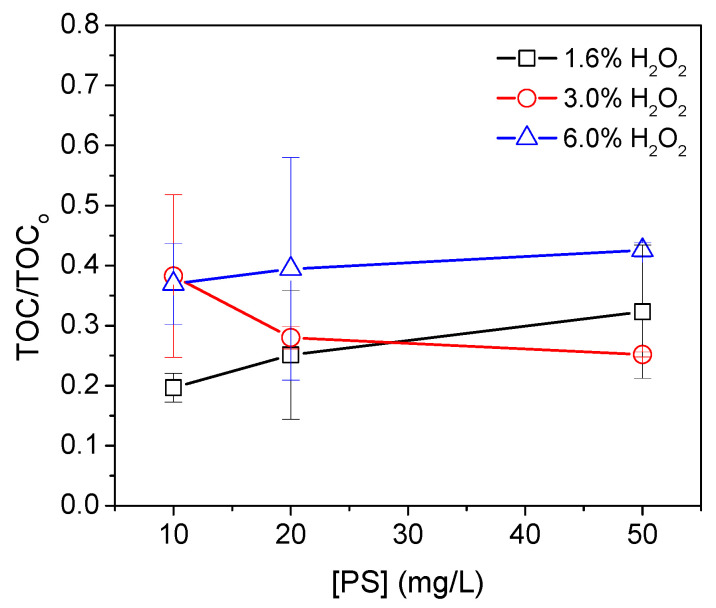
TOC ratio for different concentrations of microplastics and H_2_O_2_ concentrations. MnO_2_ 0.2 g L^−1^, triton 0.01%. pH = 9.

**Table 1 micromachines-15-00141-t001:** Comparison of Different Methodologies for Microplastic Removal.

MPs Characteristics	Material	Mechanism	Operating Conditions	Results	Reference
MPs extracted from Nivea brand facial cleanser using a 0.8 μm filter	Fe_2_O_3_-MnO_2_ with 4 μm in diameter c.a. with a spherical shape were prepared by hydrothermal reaction method.	Adsorptive bubble separation (ABS)	● [Catalyst] = 0.3 g L^−1^ ● [MPs] = 5 g L^−1^ ● 0.01% Triton-X-100 as surfactant ● 5% H_2_O_2_ ● t = 0.5–6 h	● Removal rate of 10% after 2 h of reaction	[8]
PS with a diameter 0.5–1.0 mm	Hydrophobic Fe_3_O_4_ particles were synthesized by the coprecipitation method.	Magnetic guiding forming hydrophobic twister	● 4 mg of Fe_3_O_4_ in a Petri dish with height 8.0 ± 0.5 m ● Twister speed 47.8 mm/s ● 105 mT of magnetic field	● Efficient qualitative removal for the capture of floating PS particles	[10]
PMMA sphere particles (diameter between 20 and 50 μm)	Oleic acid coated Fe_3_O_4_ particles with 10 nm in diameter c.a. were synthesized by the coprecipitation method.	Carrier flotation magnetically in duced	● [Fe_3_O_4_] = 2 g L^−1^ ● [NaCl] = 0–10 mol/L ● pH = 7	● Magnet-induced aggregation that can be collected with the assistance of permanent magnets	[11]
PE ≤ 270 μm	Magnetic magnesium hydroxide (Mg(OH)_2_ with Fe_3_O_4_ 57–147 μm) and non-ionic polyacrylamide (PAM) were synthesized by the coprecipitation method.	Coagulation	● [Mg(OH)_2_] = 50–250 mg L^−1^ ● [Fe_3_O_4_] = 40–200 mg L^−1^ ● [PAM] = 0–5 mg L^−1^ ● [MPs] = 0.05 g L^−1^	● Removal efficiency of 87.1% when Mg^2+^:OH^–^ was 1:1 ● Removal efficiency of 87.1% when accompanied by PAM	[12]
Textile fibers obtained from commercial wet wipes with a diameter ≈ 13 μm	Sphere-like Bi_2_WO_6_ particles with 6.9 μm in diameter c.a. were prepared by hydrothermal reaction method.	Degradation	● [Bi_2_WO_6_] = 1 g L^−1^ ● A piece of wipe was added to the essays ● 300 W high-pressure UV-vis lamp ● t = 50 h	● Partial degradation (unquantified)	[13]
PS beads (diameter 1 mm and 10 mm)	Polyoxometalate ionic liquid adsorbed onto magnetic microporous core–shell Fe_2_O_3_/SiO_2_ particles (magPOM-SILP) were prepared by W/O microemulsion method.	Removing by surface-binding	● [magPOM-SILP] = 10 g L^−1^ ● [MPs] = 1 g L^−1^ ● t = 24 h	● Removal efficiencies over 90%	[14]
PS spheres with 100 μm or 40 μm in diameter, and MP extracted from face cleansing cream sample	Magnetic sunflower pollen grains with 30 μm in diameter c.a. were synthesized by acidolysis and magnetic sputtering.	Shoveling	● [Catalyst] Unspecified ● [MPs] = 40–80 pieces/11–15 μL	● Removal effectiveness of 75% for the microplastics obtained from the facial cream and 70% for the PS microplastics	[15]
Cosmetic microplastics obtained by commercial facial cleansers (0.01–1.5 mm)	Magnetic N-doped nanocarbon springs with 3–5 mm in length and 20–40 nm in diam. were prepared by hydrothermal reaction method and acid treatment.	Degradation	● [Catalyst] = 0.2 g L^−1^ ● [MPs] = 5 g L^−1^ ● [PMS] = 6.5 mM ● T = 100–160 °C ● t = 0–8 h	● Activation of peroxymonosulfate (PMS) to evolve reactive radicals reaching 50% weight loss	[16]
Carboxylated PS Bead with 3 μm in diameter	Photocatalytic Au@Ni@TiO_2_ (<1 μm in diameter) chains were prepared by hydrolysis and condensation reaction.	Phoretic interaction	● [Au@Ni@TiO_2_] = 2.5 g L^−1^ ● [MPs] = 1 g L^−1^ ● 0.10–1.67% H_2_O_2_	● Removal of 77% after 120 s of reaction with 0.10% H_2_O_2_ and 63 mW UV ligh	[17]

**Table 2 micromachines-15-00141-t002:** Parameters extracted from the adsorption/desorption isotherm curves.

Sample	S_BET_ (m^2^/g)	S_MP_ (m^2^/g)	S_EXT_ (m^2^/g)	V_MP_ (cm^3^/g)	V_T_ (cm^3^/g)	Mesoporous Size (nm)	Microporous Size (nm)
δ-MnO_2_ S80	57	23	34	0.011	0.160	9.3	1.1
α-MnO_2_ S150	17	4	13	0.005	0.049	-	1.1
β-MnO_2_ S210	13	4	9	0.003	0.011	11.7	1.1, 1.4

## Data Availability

The data presented in this study are available from the corresponding author upon reasonable request.

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
