# Peer review of "Micromotors of MnO2 for the Recovery of Microplastics"

_micromachines, 2024, doi:10.3390/mi15010141_

Round 1

Reviewer 1 Report

Comments and Suggestions for Authors

This paper deals with the preparation of manganese dioxide as catalyst for microplastics removal. This work is not so interesting. There are some aspects that need to be considered. The first one is the application of this system in the real world. How do authors think to extend this study in ocean/sea? In my humble opinion it is difficult to imagine to use MnO2 and H2O2  to remove  microplastics in these aqueous enviroments. Please, insert some comments.
pag. 8 Line 208. Authors refer to some results that are not reported in the paper. Please add them.

Please, insert nitrogen isotherms.

Supplementary materials were not available.

Comments on the Quality of English Language

Fine

Author Response

Responses have been sent via attach

Reviewer 2 Report

Comments and Suggestions for Authors

The authors synthesized MnO2 as catalyst to remove microplastics from water at various conditions. I have the following comments to the manuscript.

General comments:

In lines 69-70 + Eq. 1, the authors state that MnO2 catalyses the decomposition of H2O2 to H2O and O2. Typically, the catalysts causes the H2O2 to be split into a hydroxyl radical and a hydroxyl ion. Reference 8 does not state anything about H2O and O2 formation from H2O2 using MnO2?

How is the crystallinity and surface area of the produced MnO2 compared to literature? In the methodogical section, references are provided to the synthesis, so this should be compared. Also, I suggest to add Rietveld refinements to discuss the crystallinity, crystal size etc.

Why did the authors not show any proof of the superior performance of S80 regarding removal of microplastics? And under which conditions did S80 perform better than S150 and S180? 

Similarly to the above comment, why not show the difference in pH? It is much more interesting than showing no effect of MnO2 concentration (Fig. 3). And why did the MnO2 concentration not show any difference? I would assume that the total removal might not be affected as the H2O2 concentration is the key factor for producing bubbles, however, the time to reach a certain removal level should be affected? Did the authors investigate the kinetics?

Why was a MnO2 concentration of 0.2 mg/L chosen to test other parameters? Why not use the lowest concentration necessary?

In line 232-233, the authors state that the H2O2 left in solution increases with H2O2 concentration. But how did the H2O2 concentration affect the TOC?

A conclusion section should be made to clearly show the conclusion of the work.

Where and how would the authors introduce this process in industry? What is the environmental issues with increasing the pH to 9 in order to remove microplastics? Does the surfactant cause any potential problems for the aquatic environment?

Specific comments

Supporting information is not available?

Please add a reference to the size ranges of plastics (lines 41-45).

I suggest to rotate Table 1 to ease the reading.

In line 100, I assume "dissolution" should be "solution".

Lines 131-132 should be moved to Section 2.1.

In line 148, the numbering of the section should be 2.3

In the Figure 2 (XRD), a description of the vertical lines should be added either in the figure caption or in the figure. 

Comments on the Quality of English Language

Typos should be corrected.

Author Response

Responses have been sent via attach

Round 2

Reviewer 1 Report

Comments and Suggestions for Authors

I read authors’reply. I think that comments about possible application of this system in real environment should be introduced in the introduction. Moreover, comments about results not reported in the paper should be  introduced.

Comments on the Quality of English Language

Some minor mistakes

Reviewer 2 Report

Comments and Suggestions for Authors

The authors generally improved the manuscript. I have a couple of suggestions to support some statements regarding effect of different parameters.

In my opinion, a proof of the better performance of S80 compared to S150 and S210 should be shown. The authors show no evidence, no figure or data is provided to support their statement.

The authors added a new Fig. S4 showing foam formation at different H2O2 concentration. I think it would be great to add similar pictures to support the statements that no foam formation occured, e.g., for pH effect.

Round 3

Reviewer 1 Report

Comments and Suggestions for Authors

I just have one remark:

pag 9 line 240-241 the sentence has no meaning. 

Author Response

thanks to the referee for his/her comment, indeed the sentence did not make sense, we have modified it.